# Prevalence and risk factors of disability and anxiety in a retrospective cohort of 432 survivors of Coronavirus Disease-2019 (Covid-19) from China

**Siyi Zhu**[1,2☯], **Qiang Gao**[1,2☯], **Lin Yang**[1,2☯], **Yonghong Yang**[1,2☯], **Wenguang Xia**[3], **Xiguo Cai**[4], **Yanping Hui**[5], **Di Zhu**[6], **Yanyan Zhang**[7], **Guiqing Zhang**[8], **Shuang Wu**[9], **Yiliang Wang**[10], **Zhiqiang Zhou**[11], **Hongfei Liu**[12], **Changjie Zhang**[13], **Bo Zhang**[14], **Jianrong Yang**[15], **Mei Feng**[16], **Zhong Ni**[16], **Baoyu Chen**[1], **Chunping Du**[1,2], **Hongchen He**[1,2], **Yun Qu**[1,2], **Quan Wei**[1,2], **Chengqi He**[1,2,17]\*, **Jan D. Reinhardt**[17,18,19,20☯]\*

1 Rehabilitation Medical Center, West China Hospital, Sichuan University, Chengdu, China, 2 Rehabilitation Medicine Key Laboratory of Sichuan Province, Sichuan University, Chengdu, China, 3 Hubei Hospital of Integrated Traditional Chinese and Western Medicine, Wuhan, Hubei, China, 4 Henan Provincial People's Hospital, Zhengzhou, Henan, China, 5 The Second Affiliated Hospital, Medical College of Xi'an Jiaotong University, Xian, Shanxi, China, 6 Zhejiang Provincial People's Hospital, Hangzhou, Zhejiang, China, 7 Qilu Hospital of Shandong University, Jinan, Shandong, China, 8 The First Affiliated Hospital, School of Medicine, Shihezi University, Xinjiang, China, 9 The Affiliated Hospital of Guizhou Medical University, Guiyang, Guizhou, China, 10 Chongqing Three Gorges Central Hospital, Chongqing, China, 11 Inner Mongolia Autonomous Region Baotou Central Hospital, Baotou, China, 12 General Hospital of General Bureau of Agricultural Reclamation of Heilongjiang Province, Harbin, China, 13 The Second Xiangya Hospital of Central South University, Changsha, China, 14 Nanchong Central Hospital, Nanchong, Sichuan, China, 15 Garze People's Hospital, Garze Tibetan Autonomous Prefecture, Sichuan, China, 16 Department of Critical Care Medicine, West China Hospital, Sichuan University, Chengdu, China, 17 Institute for Disaster Management and Reconstruction of Sichuan University and Hong Kong Polytechnic University, Sichuan University, Chengdu, China, 18 Swiss Paraplegic Research, Nottwil, Switzerland, 19 Department of Health Sciences and Medicine, University of Lucerne, Luzern, Switzerland, 20 Xidian Hospital Group, Xi'an, Shaanxi, China

☯ These authors contributed equally to this work.
\* hxkfhcq@126.com (CH); jan.reinhardt@paraplegie.ch (JDR)

**Data Availability Statement:** All dataset files are available from the Mendeley Data (Zhu, Siyi; Reinhardt, Jan D.; He, Chengqi (2020), "A retrospective cohort of 432 survivors of

## Abstract

### Objective

To estimate the prevalence of disability and anxiety in Covid-19 survivors at discharge from hospital and analyze relative risk by exposures.

### Design

Multi-center retrospective cohort study.

### Setting

Twenty-eight hospitals located in eight provinces of China.

### Methods

A total of 432 survivors with laboratory-confirmed SARS CoV-2 infection participated in this study. At discharge, we assessed instrumental activities of daily living (IADL) with Lawton's

Coronavirus Disease-2019 (Covid-19) from China", Mendeley Data, V2, doi: 10.17632/gwpnj5hph3.2).

**Funding:** This study was supported by the National Natural Science Foundation (81972146 to CH; 82002393 to SZ), the Department of Science and Technology of Sichuan Province (20YYJC3320 to CH; 21ZDYF1918/21YYJC2860 to SZ), China Postdoctoral Science Foundation (2020M673251), Health Commission of Sichuan Province (20PJ034), Xi'an Bureau of Technology to JDR (XA2020-HWYZ-0043), the Fundamental Research Funds for the Central Universities, China to JDR (20827041D4161), and West China Hospital of Sichuan University (HX-2019-nCoV-011 to CH; 2019HXBH058 to SZ). The funders had no role in study design, data collection and analysis, decision to publish, or preparation of the manuscript.

**Competing interests:** The authors have declared that no competing interests exist.

IADL scale, dependence in activities of daily living (ADL) with the Barthel Index, and anxiety with Zung's self-reported anxiety scale. Exposures included comorbidity, smoking, setting (Hubei vs. others), disease severity, symptoms, and length of hospital stay. Other risk factors considered were age, gender, and ethnicity (Han vs. Tibetan).

## Results

Prevalence of at least one IADL problem was 36.81% (95% CI: 32.39–41.46). ADL dependence was present in 16.44% (95% CI: 13.23–20.23) and 28.70% (95% CI: 24.63–33.15) were screened positive for clinical anxiety. Adjusted risk ratio (RR) of IADL limitations (RR 2.48, 95% CI: 1.80–3.40), ADL dependence (RR 2.07, 95% CI 1.15–3.76), and probable clinical anxiety (RR 2.53, 95% CI 1.69–3.79) were consistently elevated in survivors with severe Covid-19. Age was an additional independent risk factor for IADL limitations and ADL dependence; and setting (Hubei) for IADL limitations and anxiety. Tibetan ethnicity was a protective factor for anxiety but a risk factor for IADL limitations.

## Conclusion

A significant proportion of Covid-19 survivors had disability and anxiety at discharge from hospital. Health systems need to be prepared for an additional burden resulting from rehabilitation needs of Covid-19 survivors.

## Introduction

With rapidly increasing numbers of confirmed cases and deaths caused by the Coronavirus Disease-2019 (Covid-19) pandemic, clinical outcomes research has almost exclusively focused on disease progression and mortality [1]. With the case fatality rate estimated to lie somewhere between 0.56 and 9.38 percent globally (95% prediction interval, $I^2 = 100\%$) [2], it is obvious however that a vast majority of people actually survive Covid-19. A substantial proportion of survivors have experienced severe disease episodes requiring hospitalization and—in a considerable number of cases—intensive care including mechanical ventilation [3]. In addition, contracting a potentially fatal disease is stressful and can cause anxiety and other psychiatric manifestations in many patients [4]. It would be naïve to assume that health problems just disappear upon discharge from acute care or when viral RNA is no longer detectable, and equally naïve to take planned and systematic follow up of patients for granted [5, 6]. Yet, the burden of disease resulting from physical and psychological sequelae of Covid-19 in patients surviving the disease as well as potential long-term adverse effects from treatments including steroids such as Dexamethasone are just beginning to catch attention [7, 8].

Globally, respiratory infections and tuberculosis assume already the fourth rank of all diseases in terms of disability adjusted life years (DALYs) and the 17[th] rank in years lived with disability (YLD). In infectious disease, they sadly rank first and second, respectively [9, 10]. Covid-19 will increase this burden considerably and cause longer term mental and physical health problems, work disability, and reduced quality of life in survivors, frontline health professionals, and those quarantined [11, 12]. What we know from the 2002/2003 Severe Acute Respiratory Syndrome (SARS) epidemic is not encouraging. In Hong Kong for example, about 80 percent of SARS survivors still needed follow-up treatment two years after the outbreak had been contained [13].

The number of patients needing follow-up care and rehabilitation due to Covid-19 will be unprecedented. Countries need to consider and plan for this additional long-term challenge to the health system. It is important to understand risk factors for disability and mental health problems now in order to direct scarce resources and meet windows of opportunity for treatment. This study makes a first attempt to address this issue. We aimed to (1) estimate the prevalence of disability and anxiety in Covid-19 survivors from eight Provinces/centrally governed municipalities of the PR China at discharge from acute inpatient treatment, and (2) investigate relative risk of adverse outcomes by various determinants including gender, age, comorbidity, setting, ethnicity and disease severity. Prevalence estimates are important to gain insight in the potential number of patients needing follow-up care. Analyzing risk factors is instrumental in identifying vulnerable groups and allocating resources for early interventions.

## Methods

### Study design

We conducted a multi-center retrospective cohort study.

### Ethics procedures

The study protocol was approved by the ethics committee of West China Hospital, Sichuan University (2020–163 and 2020–273). All participating health professionals were instructed on implementation of the study protocol, informed consent procedures, and possible sources of bias by videoconference. Patients were informed by the health professionals either face to face (if they were still hospitalized) or via telephone or WeChat call (if they had already been discharged) about the study purpose and content, and that participation was voluntary and could be withdrawn at any time. Informed consent was then obtained either in written form or verbally. Verbal informed consent was obtained if the patients were either illiterate or had already been discharged. Verbally provided informed consent was witnessed by a second health professional and recorded electronically. The data analyzed and contact information for the participants were strictly separated with the only link between them being a unique ID number without identifying information. Only Drs S Zhu and C He had access to both datasets.

### Study participants

Four hundred and thirty-two Covid-19 survivors were surveyed who had received treatment in 28 designated hospitals in Hubei, Sichuan, Guizhou, Henan, Neimenggu (inner Mongolia), Shandong, Hainan, and Chongqing with admission dates from January 18 to March 15, 2020 and gave informed consent. Included were adults ($\geq$ 16 years) who had laboratory-confirmed SARS coronavirus-2 (SARS CoV-2) infection and met the diagnostic criteria of the Novel Coronavirus Pneumonia Diagnosis and Treatment Program (Trial Version 7) issued by the National Health Commission of the PR China (Third edition) [14]. Survivors further needed to be in stable medical condition and have been discharged or be about to be discharged from the participating centers. Excluded were patients with pneumonia caused by other types of coronaviruses, seasonal influenza, bacteria, or other not SARS CoV-2 etiology.

### Measures

Data on outcomes and patient-reported exposures/risk factors was collected and informed consent was obtained between February 21 and April 7, 2020.

**Outcomes.** Outcomes were disability and anxiety. Cut off date for outcome assessment was April 7, 2020. Disability was measured in two ways: (1) Limitations in instrumental

activities of daily living (IADL) were assessed with the Lawton IADL scale [15] and classified into no limitations vs. one or more limitations with the latter indicating an unfavorable primary outcome. An index of the number of reported IADL limitations was also created. (2) Dependence in activities of daily living (ADL) was measured with the Barthel Index [16] and classified into severe or moderate dependence vs. mild dependence/independence with BI scores smaller than 75 indicating an unfavorable secondary outcome [17]. Anxiety was evaluated with Zung's Self-Reported Anxiety Scale [18] (SAS) and classified into probable clinical anxiety disorder vs. not according to a conservative cutoff recommended for research with SAS scores of 40 or greater indicating an unfavorable secondary outcome [19]. Outcome data were patient-reported and collected with an online questionnaire.

**Risk factors.** Clinical and demographic data were extracted from hospital records by participating health professionals. Current smoking status was self-reported by participants. Demographic data included age, gender, province (Hubei vs. others), and ethnicity (Han vs. Tibetan). Having received treatment in Hubei was considered a risk factor because of the particular situation in Hubei which was the epicenter of the epidemic in China representing about 80 percent of confirmed cases. Ethnicity was included as a significant proportion of the Sichuan Province sample belonged to the Tibetan ethnic group, and belonging to a minority ethnic group is known to be associated with health outcomes in other disease. Clinical data included comorbidity, disease severity, symptoms at admission, scope of pneumonia, and length of inpatient stay. Recorded comorbid conditions included COPD, chronic bronchitis, pulmonary fibrosis, hypertension, diabetes, cardiovascular disease, kidney disease, hepatitis, rheumatism, and gout. Degree of Covid-19 severity was classified according to the Chinese standard [14]. Patients were defined as severe cases when they met one of the following criteria at any time during hospitalization: acute respiratory distress, respiratory rate $\geq$30 breath/min; pulse oxygen saturation (SpO2) $\leq$93% at rest; arterial blood partial pressure of oxygen/fraction of inspired oxygen (PaO2/FiO2) $\leq$300 mmHg (1 mmHg = 0.133 kPa); respiratory failure requiring mechanical ventilation; septic shock; failure of other organs requiring ICU treatment. Symptoms recorded at admission included fever, cough, diarrhea, fatigue and pain. Scope of pneumonia was noted as uni- or bilateral per radiographic findings. Data on length of hospital stay were available for 408 patients (94.44%).

## Sample size calculation

Sample size to detect a two-fold relative risk for reporting an unfavorable primary outcome, i.e. one or more IADL limitations, in the severe disease group as compared to the non-severe group with a power of 80% and alpha error of 5% was estimated under the following assumptions: 0.2 ratio of severe to non severe group (based on patients received at our own center), 15 percent prevalence of outcome in non-severe group. This yielded a minimal sample size of 401 (67 severe, 334 non-severe).

## Analysis

Demographic and clinical characteristics of the sample are described by providing numbers of participants and percentages (categorical variables) or medians and inter-quartiles (continuous variables). Non-severe and severe cases are compared with regard to these characteristics with p-values for differences estimated from chi-squared tests (categorical variables) or Mann-Whitney U tests (continuous variables). Prevalence of limitations in IADL, ADL dependence, and anxiety for the full sample and by disease severity are provided with 95% confidence intervals (CIs) and p-values estimated from logistic regression. The strength of association between outcomes is reported as Cramer's V with p-values from chi-squared tests. Cramer's V ranges

from zero (no association) to one (perfect association) with values from 0.3 to 0.5 indicating moderate effects and values above 0.5 indicating large effects [20]. Risk ratios (RR) and 95% CIs for unfavorable outcomes (disability, anxiety) by gender, age group ($< = 50$, 50–60, $> 60$ years), province (others vs. Hubei), ethnicity (Han vs. Tibetan), comorbid health conditions (none, one, multiple), disease severity (non-severe vs. severe), symptoms ($< = 1$, $>1$) at admission, and infection scope (unilateral vs. bilateral) were estimated with log-linear Poisson regression with robust standard errors [21]. Since longer time of being hospitalized could increase anxiety, length of hospital stay was included in the anxiety-model in addition. For each outcome unadjusted RR and RR adjusted for all other potential risk factors evaluated are provided. Zero-inflated Poisson (ZIP) regression was used to estimate the effect of risk factors on the extent of IADL limitations. ZIP regression assumes that there are two groups in the population from which the data are sampled, one which always has zero counts, and one which may have zero or higher counts. Young people with non-severe disease who have no comorbidity may never develop IADL limitations, while other groups may develop (additional) IADL limitations due to factors associated with disease progression. ZIP regression simultaneously performs logistic and Poisson regression making two types of predictions: (1) prediction of excess zeros assuming a binary dependent variable (no IADL limitations vs. one or more), (2) prediction of the count portion (number of experienced IADL limitations). Predictors included in the logistic part were the following background factors: age group and interaction of age and disease severity, province, ethnicity, smoking, and comorbid conditions. Predictors included in the count part were potential drivers of disease consequences: disease severity, interaction of age and disease severity, symptoms, and infection scope. Vuong's test confirmed fit of the ZIP model.

All analyses were performed with Stata 14 (Stata corporation, Texas, USA).

## Results

### Sample description and prevalence of outcomes

Demographic and clinical characteristics and prevalence of outcomes of the study participants are provided in Table 1. Median age was 49 (IQR 35–60) and 49% were female. About one third of the study participants were classified as severe cases. Fever, cough, and fatigue were the most common symptoms at admission. A majority had no pre-existing health condition and were non-smokers. Patients with severe Covid-19 were older, were more often from Hubei and belonged to the Han majority, presented with more symptoms, had more often bilateral pneumonia, and stayed in the hospital for a longer time (all p<0.001). More pre-existing health conditions were present in severe cases (p<0.001), but the percentage of smokers was lower (p = 0.140). Over one third of Covid-19 survivors had at least one IADL limitation, about 15 percent had at least moderate ADL dependence, and six percent were severely dependent. Probable clinical anxiety disorder was found in about 29 percent of the population. Prevalence of all outcomes was more than four times higher in survivors of severe Covid-19 than in those with non-severe disease (all p < 0.001).

### Associations between outcomes

Outcomes showed moderate to strong associations with Cramer's V being 0.56 for the association between one or more IADL limitation and at least moderate dependence, 0.39 for the association between one or more IADL limitation and probable clinically relevant anxiety, and 0.33 for the association between at least moderate dependence and anxiety (all p < 0.001).

Table 1. Demographic and clinical characteristics of study participants (I) and prevalence of outcomes (II).

| I. Demographic and clinical characteristics | All participants (N = 432) | By disease severity | | |
|---|---|---|---|---|
| | | non-severe | severe | P |
| | | (n = 285, 65.97%) | (n = 147, 34.03%) | |
| **Female sex–no./total no. (%)** | 207/432 (47.92) | 138/285 (48.42) | 69/147 (46.94) | 0.770[+] |
| **Age** | | | | |
| Median (IQR) in years | 49 (35–60) | 45 (32.5–54.5) | 57 (47–68) | <0.001[‡] |
| Distribution–no./total no. (%) | | | | <0.001[+] |
| < 50 | 218/431 (50.58) | 173/284 (60.92) | 45/147 (30.61) | |
| 50–60 | 108/431 (25.06) | 69/284 (24.30) | 39/147 (26.53) | |
| > 60 | 105/431 (24.36) | 42/284 (14.79) | 63/147 (42.86) | |
| **Province–no./total no. (%)** | | | | <0.001[+] |
| Hubei | 169/432 (39.12) | 73/285 (25.61) | 96/147 (65.31) | |
| Sichuan | 162/432 (37.50) | 144/285 (50.53) | 18/147 (12.24) | |
| Chongqing | 38/432 (8.80) | 29/285 (10.18) | 9/147 (6.12) | |
| Henan | 30/432 (6.94) | 15/285 (5.26) | 15/147 (10.20) | |
| Guizhou | 24/432 (5.56) | 17/285 (5.96) | 7/147 (4.76) | |
| Other | 9/432 (2.08) | 7/285 (2.46) | 2/147 (1.36) | |
| **Ethnicity–no./total no. (%)** | | | | <0.001[+] |
| Han | 360/432 (83.33) | 220/285 (77.19) | 140/147 (95.24) | |
| Tibetan | 72/432 (16.67) | 65/285 (22.81) | 7/147 (4.76) | |
| **Smoking history, yes–no./total no. (%)** | 62/432 (14.35) | 46/285 (16.14) | 16/147 (10.88) | 0.140[+] |
| **Pre-existing comorbidity–no./total no. (%)** | | | | <0.001[+] |
| None | 300/432 (69.44) | 226/285 (79.30) | 74/147 (50.34) | |
| One | 79/432 (18.29) | 37/285 (12.98) | 42/147 (28.57) | |
| Multi | 53/432 (12.27) | 22/285 (7.72) | 31/147 (21.09) | |
| **Symptoms on admission–no./total no. (%)** | | | | |
| Fever | 249/432 (57.64) | 125/285 (43.86) | 124/147 (84.35) | <0.001[+] |
| Cough | 257/432 (59.49) | 139/285 (48.77) | 118/147 (80.27) | <0.001[+] |
| Fatigue | 153/432 (35.42) | 70/285 (24.56) | 83/147 (56.46) | <0.001[+] |
| Pain | 68/432 (15.74) | 40/285 (14.04) | 28/147 (19.05) | 0.175[+] |
| Diarrhea | 46/432 (10.65) | 24/285 (8.42) | 22/147 (14.97) | 0.037[+] |
| **Bilateral infection scope[&] - no./total no. (%)** | 362/432 (83.80) | 221/285 (77.54) | 141/147 (95.92) | <0.001[+] |
| **Length of hospital stay[§]** | | | | |
| Median (IQR) in days | 18 (12–25) | 15 (11–22) | 23 (15.5–31.61) | <0.001[‡] |
| Distribution–no./total no. (%) | | | | <0.001[+] |
| 1–14 days | 160/408 (39.22) | 130/268 (48.51) | 30/140 (21.43) | |
| >14 days | 248/408 (60.78) | 138/268 (51.49) | 110/140 (78.57) | |
| **Discharge Destination[§] - no./total no. (%)** | | | | <0.001[+] |
| Discharged home | 278/412 (67.48) | 215/269 (79.93) | 63/143 (44.06) | |
| Discharged to 14-day quarantine | 128/412 (31.07) | 52/269 (19.33) | 76/143 (53.15) | |
| Referred to other hospital | 6/412 (1.46) | 2/269 (0.74) | 4/143 (2.80) | |
| **Mechanical Ventilation–no./total no. (%)** | | | | <0.001[+] |
| No | 395/411 (96.11) | 268/268 (100.00) | 127/143 (88.81) | |
| Yes | 16/411 (3.89) | 0/268 (0.00) | 16/143 (11.19) | |
| **II. Outcomes[*]** | | | | |
| **Disability, percent (95% CI)** | | | | |
| One or more IADL limitations | 36.81 (32.39–41.46) | 18.25 (14.18–23.16) | 72.79 (65.04–79.37) | <0.001[*] |
| Moderate ADL dependence | 16.44 (13.23–20.23) | 6.32 (4.01–9.80) | 36.05 (28.71–44.11) | <0.001[*] |

*(Continued)*

**Table 1.** (Continued)

| I. Demographic and clinical characteristics | All participants (N = 432) | By disease severity | | |
|---|---|---|---|---|
| | | non-severe | severe | P |
| | | (n = 285, 65.97%) | (n = 147, 34.03%) | |
| Severe ADL dependence | 5.56 (3.75–8.15) | 2.46 (1.17–5.06) | 11.56 (7.31–17.82) | <0.001* |
| **Anxiety, percent (95% CI)** | 28.70 (24.63–33.15) | 13.33 (9.86–17.80) | 58.50 (50.38–66.19) | <0.001* |

§Scope of pneumonia as confirmed by radiography.

&For 24 (5.56%) patients discharge date had not been entered in the data form.

+P-values from chi-squared test.

‡ P-values from Mann-Whitney U test.

*Confidence intervals and p-values estimated from logistic regression.

CI = confidence interval. IADL = instrumental activities of daily living. ADL = activities of daily living.

## Effects of risk factors on the occurrence of adverse outcomes

Results on risk factors for disability and anxiety are displayed in Figs 1 and 2 (for detailed estimates see S1-S3 Tables in S1 Appendix). Risk ratios for reporting at least one IADL limitation (Fig 1, upper panel) were increased in the older age groups, particularly in survivors aged older than 60 years (adjusted RR 2.518, 95% CI 1.800–3.525), in patients from Hubei (adjusted RR 2.582, 95% CI 1.810–3.684), and in survivors with severe Covid-19 (adjusted RR 2.476, 95% CI 1.801–3.404). The adjusted model also showed an elevated relative risk in survivors from the Tibetan ethnic group (adjusted RR 2.391, 95% CI 1.513–3.780). Only in unadjusted models was the relative risk of having one or more IADL limitations clearly elevated in survivors with comorbidities (unadjusted RR for one comorbidity 2.211, 95% CI 1.694–2.886, unadjusted RR for multi-comorbidity 2.436, 95% CI 1.847–3.212), in patients with more symptoms at admission (unadjusted RR 1.602 95% CI 1.217–2.110), and in those with bilateral infections (unadjusted RR 2.172, 95% CI 1.308–3.605). Both in unadjusted and adjusted models, the relative risk for ADL dependence (Fig 1 lower panel) was consistently increased in Covid-19 survivors aged older than 60 years (adjusted RR 6.783, 95% CI 3.155–14.586), and in participants with severe disease course (adjusted RR 2.074, 95% CI 1.146–3.755) and more symptoms (adjusted RR 2.074, 95% CI 1.129–3.810). All individual risk ratios were higher in unadjusted models. Setting (unadjusted RR 3.048, 95% CI 1.938–4.792), comorbidity (unadjusted RR for one comorbidity 3.505, 95% CI 2.133–5.762; unadjusted RR for multi-comorbidity 4.572, 95% CI 2.784–7.507), and infection site (unadjusted RR 2.095, 95% CI 0.944–4.649) played a role in unadjusted analysis only.

Having severe Covid-19 (adjusted RR 2.533, 95% CI 1.693–3.788) was the strongest risk factor for probable clinically relevant anxiety (Fig 2), followed by having received treatment in Hubei province (adjusted RR 2.055, 95% CI 1.422–2.972). In turn, belonging to the Tibetan group was associated with a largely decreased relative risk of anxiety (adjusted RR 0.214, 95% CI 0.050–0.916). There was a trend for an increased relative risk of anxiety in survivors who had stayed in the hospital for more than 14 days (adjusted RR 1.482, 95% CI 0.998–2.200). Older age (unadjusted RR for being older than 60 2.318, 95% CI 1.650–3.255), comorbidity (unadjusted RR for one comorbidity 1.899, 95% CI 1.345–2.681; unadjusted RR for multi-comorbidity 2.476, 95% CI 1.772–3.462), symptoms (unadjusted RR 2.303, 95% CI 1.600–3.315), and scope of pneumonia (unadjusted RR 3.232, 95% CI 1.574–6.635) played a role in unadjusted analysis only.

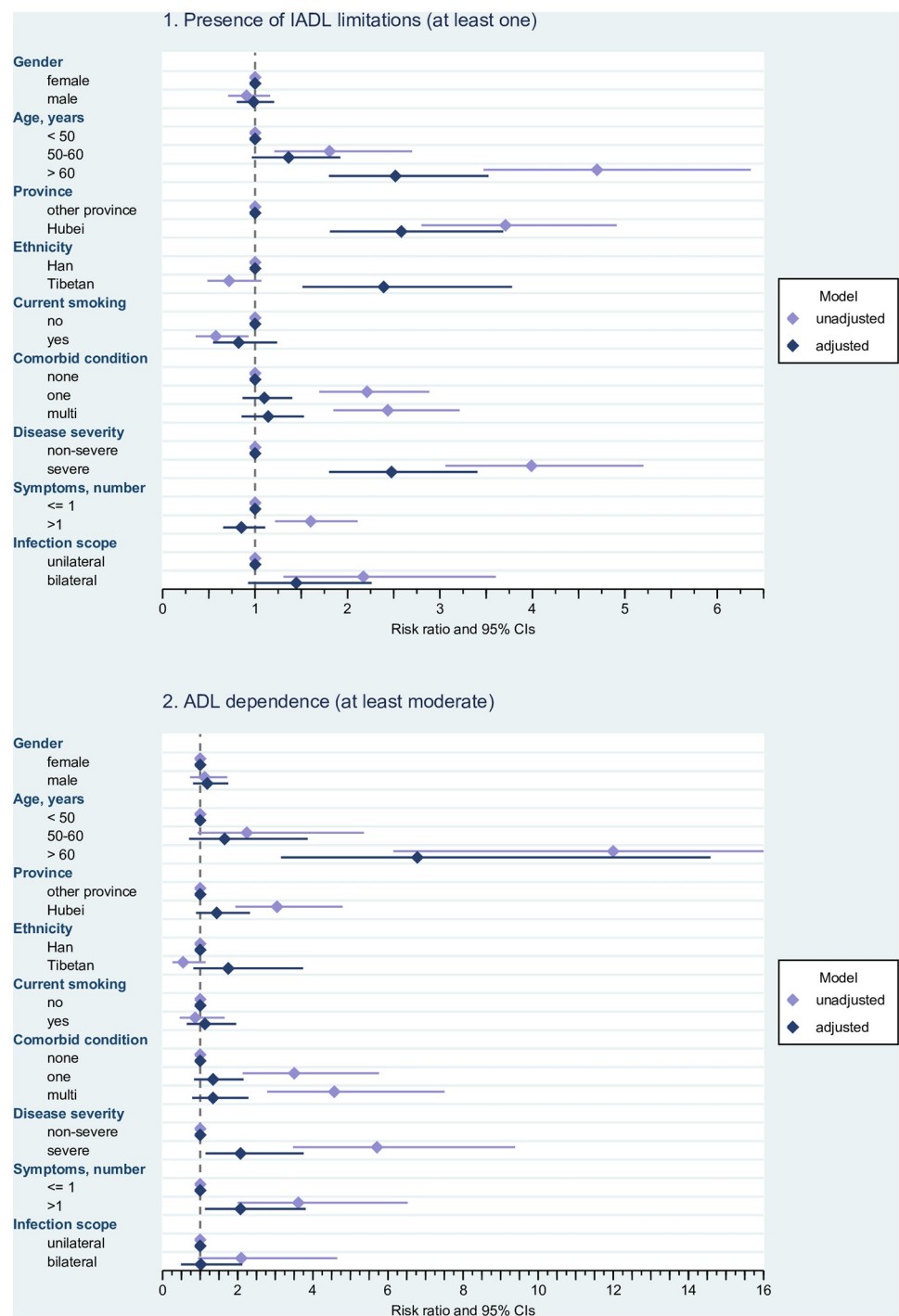

**Fig 1. Disability (IADL limitations and ADL dependence) in relation to potential risk factors.** Risk ratios and 95% confidence intervals. Estimated from log-linear Poisson regression with robust standard errors; unadjusted risk ratios are estimated from univariable models for the effect of the respective predictor on the outcome, adjusted risk ratios are estimated from a multivariable model containing all potential risk factors. IADL = Instrumental Activities of Daily Living. ADL = Activities of Daily Living.CI = confidence interval. Note: In the case of ADL dependence, the upper limit of the confidence interval for the unadjusted estimate of age > 60 years has been truncated for better readability of the other effects (see S1-S3 Tables in S1 Appendix for details). Adjusted model and unadjusted model for age have been estimated for 431 cases (because of one missing date of birth).

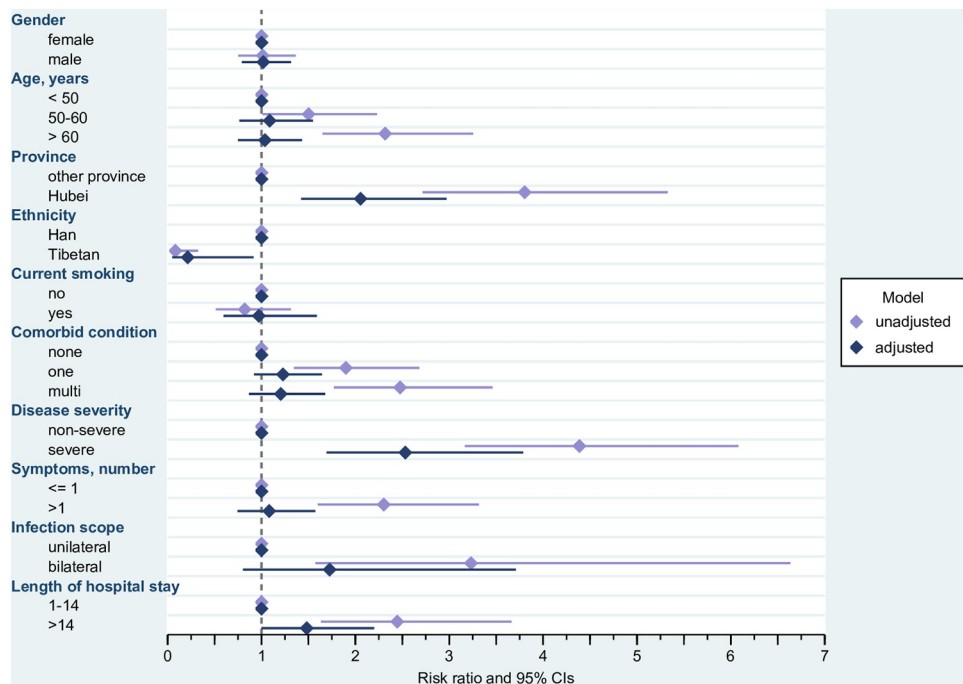

**Fig 2. Probable clinical anxiety in relation to potential risk factors.** Risk Ratios and 95% confidence intervals. Estimated from log-linear Poisson Regression with robust standard errors; unadjusted risk ratios are estimated from univariable models for the effect of the respective predictor on the outcome, adjusted risk ratios are estimated from a multivariable model containing all potential risk factors. The adjusted model has been estimated for 407 cases (because of missing information regarding length of stay for 24 cases and one case with missing date of birth). [&]Unadjusted model estimated for 407 cases (because of missing information regarding length of stay for 24 cases and one case with missing date of birth). CI = confidence interval.

## Effects of risk factors on the number of limitations in IADL

Table 2 show the results of the ZIP regression of the number of IADL limitations on risk factors. Hospitalization in Hubei, Tibetan ethnicity (IRR 0.16, 95% CI 0.06–0.41), and multi-comorbidity (IRR 0.33, 95% CI 0.12–0.90) were associated with lower odds of having a zero count of IADL limitations. For all age groups, odds for having zero IADL limitations were decreased in people with severe Covid-19. This effect became however smaller with increasing age. IRRs were 0.06 (95% CI 0.02–0.17) for the youngest age group with severe Covid-19, 0.11 (95% CI 0.04–0.32) for the middle age group, and 0.34 (95% CI 0.10–1.16) for the oldest participants with severe Covid-19 as compared to those with non-severe disease. Disease severity (IRR 1.64, 95% CI 0.97–2.80), its interaction with age (see Table 2 for detailed estimates), and symptoms (IRR 1.34, 95% CI 1.10–1.62) were associated with a greater number of health problems. The joint effect of age and degree of severity of Covid-19 on the number of IADL limitations from both parts of the model is illustrated in Fig 3. In all age groups having a severe course of Covid-19 was associated with a higher number of self-reported IADL limitations. Point estimates for survivors with non-severe and those with severe disease were however closer in the oldest age group. CIs for the number of IADL limitations in participants with non-severe and those with severe Covid-19 moved closer in both older age groups and overlapped in survivors aged 61 years and older.

**Table 2. Results from zero-inflated Poisson regression of number of IADL limitations on potential risk factors (n = 431).**

| I. Excess zeros: Prediction of having no IADL limitation | OR (95% CI) | SE | z | P |
|---|---|---|---|---|
| **Age** | | | | |
| < 50 | 1 (reference) | | | |
| 50–60 | 0.66 (0.24–1.78) | 0.51 | -0.82 | 0.410 |
| > 60 | 0.04 (0.01–0.10) | 0.54 | -6.19 | <0.001 |
| **Disease severity and age group** | | | | |
| non-severe and < 50 years old | 1 (reference) | | | |
| severe and < 50 years old | 0.06 (0.02–0.17) | 0.51 | -5.43 | <0.001 |
| non-severe and 50–60 years old | 1 (reference) | | | |
| severe and 50–60 years old | 0.11 (0.04–0.32) | 0.56 | -4.02 | <0.001 |
| non-severe and > 60 years old | 1 (reference) | | | |
| severe and > 60 years old | 0.34 (0.10–1.16) | 0.62 | -1.72 | 0.086 |
| **Province** | | | | |
| Other provinces | 1 (reference) | | | |
| Hubei | 0.09 (0.04–0.19) | 0.37 | -6.46 | <0.001 |
| **Ethnicity** | | | | |
| Han | 1 (reference) | | | |
| Tibetan | 0.16 (0.06–0.41) | 0.49 | 3.78 | <0.001 |
| **Smoking history** | | | | |
| no | 1 (reference) | | | |
| yes | 1.78 (0.67–4.75) | 0.50 | 1.15 | 0.251 |
| **Comorbid conditions** | | | | |
| none | 1 (reference) | | | |
| one | 0.65 (0.29–1.43) | 0.41 | -1.08 | 0.281 |
| multi | 0.33 (0.12–0.90) | 0.51 | -2.16 | 0.031 |
| **II. Count: Number of IADL limitations** | IRR (95% CI) | SE | z | |
| **Disease severity (reference: non-severe)** | | | | |
| non severe | 1 (reference) | | | |
| severe | 1.64 (0.97–2.80) | 0.45 | 1.84 | 0.066 |
| **Disease severity and age group** | | | | |
| non-severe and < 50 years old | 1 (reference) | | | |
| non-severe and 50–60 years old | 1.76 (0.99–3.15) | 0.52 | 1.92 | 0.055 |
| non-severe and > 60 years old | 2.91 (1.78–4.77) | 0.73 | 4.24 | <0.001 |
| severe and < 50 years old | 1 (reference) | | | |
| severe and 50–60 years old | 1.34 (1.03–1.74) | 0.18 | 2.21 | 0.027 |
| severe and > 60 years old | 1.65 (1.33–2.05) | 0.18 | 4.52 | <0.001 |
| **Symptoms, number** | | | | |
| < = 1 | 1 (reference) | | | |
| >1 | 1.34 (1.10–1.62) | 0.13 | 2.98 | 0.003 |
| **Infection scope** | | | | |
| Unilateral | 1 (reference) | | | |
| Bilateral | 1.19 (0.89–1.61) | 0.18 | 1.16 | 0.246 |

OR = odds ratio, CI = confidence interval, SE = standard error of coefficient, z = standardized coefficient, IRR = incidence rate ratio, IADL = instrumental activities of daily living.

## Discussion

This retrospective cohort study found a considerable prevalence of functional limitations and dependence as well as anxiety in Covid-19 survivors at about time of discharge from acute

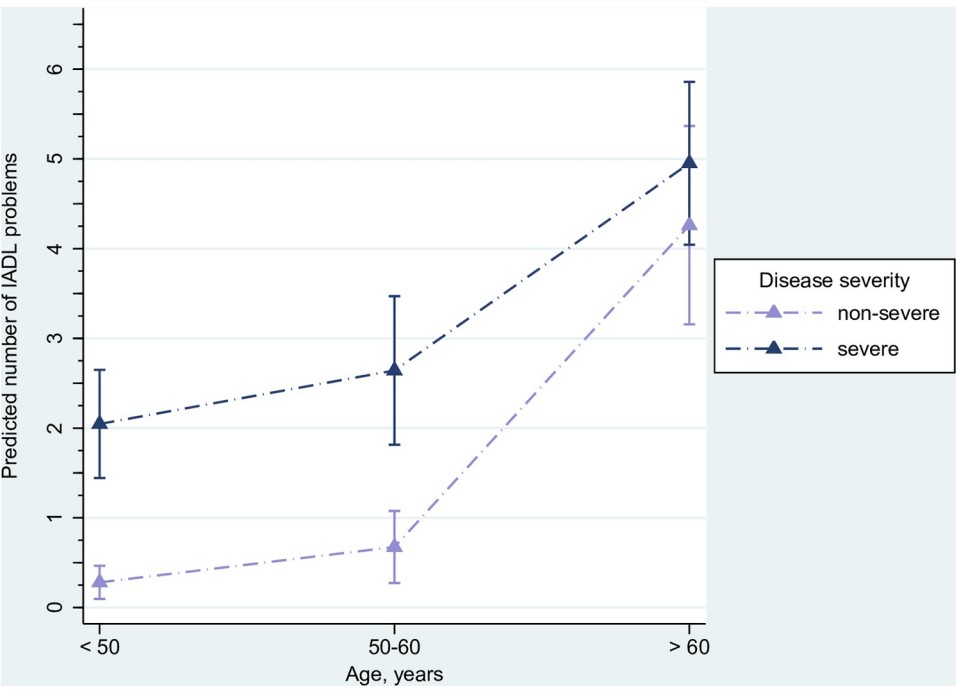

**Fig 3. Effect of the interaction of age and disease severity on the number of predicted IADL problems (triangles) with 95% confidence intervals (capped lines).** Estimated from zero-inflated Poisson regression. Predictions are adjusted for other factors in the model. Predictors included in the logistic part (excess zeros) were the following: age group, interaction of age and disease severity, setting (Hubei vs. other province), Tibetan ethnicity (reference: Han), smoking history, and comorbid conditions. Predictors included in the count part were: disease severity, interaction of age and disease severity, symptoms, and infection scope. The model has been estimated for 431 cases (because of one missing date of birth).

inpatient treatment. Disease severity was a major independent risk factor for all outcomes. Age was an additional risk factor for both disability outcomes, and setting (Hubei) for IADL limitations and anxiety. Tibetan ethnicity was a strong protective factor for anxiety but a risk factor for IADL limitations when it was adjusted for covariates.

Demographic and clinical characteristics of the study population were comparable to other reports on Covid-19 patients from China [1, 3] with two exceptions: a higher percentage was classified as severe cases and current smoking was more prevalent in the non-severe disease group. An explanation for the higher percentage of severe cases in the present study is that the applied criteria for classifying Covid-19 cases as severe by the National Health Commission of the PR China are more liberal than the criteria for severe community-acquired pneumonia by the American Thoracic Society and Infectious Disease Society of America [22] used by Guan et al. [3]. The lower prevalence of smokers in participants with severe disease was unexpected and contradicts previous evidence in this regard [3]. A possible explanation is that current smoking status was self-reported and participants with more severe disease outcomes did not reveal their true smoking status, perhaps for fear of being stigmatized and held responsible for contracting SARS CoV-2 or the severity of Covid-19.

We found a high prevalence of disability and anxiety, particularly in severe cases. Almost three quarters of severe Covid-19 survivors experienced at least one IADL problem and more than one third had at least moderate ADL dependence. The six percent of the overall sample with severe ADL dependence are of particular concern since severe functional dependence has been a reliable predictor of mortality in populations with other health conditions [23]. We

found moderate to strong associations between outcomes, alluding to possible processes of mediation, i.e. indirect effects on one outcome triggered by the other outcomes. No other original study directly investigated disability outcomes related to Covid-19 so far. Follow up studies of SARS and Middle East Respiratory Syndrome (MERS) survivors have however shown that decreased exercise capacity and distance walked, fatigue, sleeping problems, shortness of breath, reduced self-rated health, work disability, and structural damage (e.g. osteonecrosis) from corticosteroids were prevalent in survivors even years after onset of symptoms [11, 24, 25]. In addition, access to health care, quality of services, and employment opportunities of former SARS patients were compromised by enduring stigma [13, 26]. Our findings imply that a large proportion of Covid-19 survivors, particularly those who experienced severe disease episodes are in need of follow up care and rehabilitation services. This proportion may be reduced by the integration of early rehabilitation interventions into acute care [27, 28].

Probable clinical anxiety disorder was found in one third of the overall sample and its prevalence was almost four-fold increased in severe as compared to non-severe cases. Previous studies on mental health in the Covid-19 pandemic have been conducted in the general population [29, 30] and health professionals [31] but not in the patient population. Studies in SARS survivors have however reported highly elevated long-term psychiatric morbidity [32] and demonstrated adverse effects on the mental health of caregivers [11]. Our findings point to the need of psychological counseling during acute care and mental health follow-up in many Covid-19 survivors to mitigate suffering and prevent the manifestation of psychiatric disorders.

Major risk factors for mortality due to Covid-19 [1, 33] were also major risk factors of disability: older age and severe disease course both independently increased the relative risk of IADL limitations and ADL dependence. However, comorbidity [34] had an adverse effect on disability outcomes in unadjusted analysis only. Though attenuated, disease severity continued to play a major role when it was adjusted for age, co-morbidity, and other potential confounders demonstrating an additional effect of Covid-19 when other known risk factors for disability were held statistically constant. In all age groups having severe Covid-19 was associated with a higher number of self-reported IADL limitations. For survivors aged over 60 years we found that the extent of IADL limitations was more similar in those with non-severe and severe Covid-19 than in the other age groups. Accordingly, the oldest participants with non-severe Covid-19 were more similar to survivors with severe disease in the younger age groups. There are two possible explanations for this phenomenon that deserve further scrutiny: First, older people have already more activity limitations before they acquire Covid-19. Second, a non-severe disease course has a greater effect on IADL limitations for them. Persistence of this effect when it was adjusted for comorbidity speaks for the latter explanation. Having been treated in hospitals of Hubei province where Covid-19 spread first and overwhelmed an unprepared health system was furthermore associated with adverse disability outcomes, particularly IADL limitations. Moreover, belonging to the Tibetan ethnic minority predicted worse IADL outcomes when it was controlled for other potential risk factors. Given that the Tibetan population of Sichuan province mainly lives in rural, mountainous areas, both findings point at the role that environmental factors including the availability and accessibility of targeted services play in the disablement process [35]. Our findings indicate that priority in rehabilitation resource allocation should be given to patients with severe disease course, older people, and those living in environments with restricted accessibility of services.

Regarding anxiety, outstanding independent risk factors were severity of Covid-19 and setting, i.e. having been hospitalized in Hubei province. Apart from the specific situation in hospitals in Hubei, insecurity about causes, transmission patterns, and prognosis when Covid-19 first occurred likely contributed to this finding. Moreover, we found a trend for longer length

of hospital stay being independently associated with increased relative risk of anxiety. In contrast to disability outcomes, Tibetan ethnicity was a consistent protective factor in unadjusted as well as adjusted models of anxiety. Possible explanations are cultural factors including larger families (due to the non application of China's one child policy to this group) and greater social cohesion and support. This finding needs further exploration. Our results point towards heightened need for psychological support in patients who experienced severe disease episodes and people from areas hardest hit by the Covid-19 pandemic [36, 37].

## Study limitations

The present study has limitations. First, baseline data for the outcomes analyzed here were not available. Because of this we cannot rule out that patients had disability or anxiety before they contracted Covid-19, indeed having disability may have made people more vulnerable to the disease [38–40]. However, most patients first presented already with severe symptoms and baseline information collected before admission is rare in any disease. Baseline information on comorbidity is thus as close as we get to baseline disability and the effects of disease severity on outcomes remained consistent when adjusted for baseline comorbidity in our analysis. Second, we may be criticized for not having employed some kind of control group. An appropriate control group for the effects and population under investigation is however difficult to define and to establish. Age and gender matched general population controls are surely not appropriate because of the lack of hospitalization. Investigating patients who have been hospitalized for some other disease would not help disentangle the effects of disease and hospitalization. Quarantined persons suspected for Covid-19 or those treated at home may make up an interesting comparator but are difficult to access. We believe that our comparison of patients with severe and non-severe Covid-19 nonetheless provides initial evidence for the potential impact of the disease on function and mental health, and is an important starting point in investigating this population's rehabilitation needs. Third, outcome data were self-reported by patients due to the lack of options for objective measurement. Survivors thus may have under- or overestimated their physical abilities. Patient-reported outcome measures have however not been used in this population before to the best of our knowledge, making it difficult to appraise if and to what degree this kind of problem existed. Fourth, psychiatric confirmation of anxiety in those screened positive was not possible. Fifth, outcomes were correlated with each other pointing at possible mediation of effects. Indirect effects due to mediation were however not formally examined here given the cross-sectional nature of outcome assessment. Furthermore, we refrained from more complex analyses due to the novel character of the presented data and the therefore exploratory character of analysis. Sixth, this study evaluated outcomes at discharge and conclusions about longer-term prevalence, spontaneous recovery, and chronic manifestations cannot be drawn at this point and further follow-up of the cohort is therefore indicated.

## Recommendations

In spite of these limitations, there are several recommendations that can be derived from our findings. First, awareness about possible functional and psychological consequences of Covid-19 needs to be raised in patients, care providers, and health policy makers [28]. Second, early psychological [41] and pulmonary rehabilitation interventions including mobilization and exercise [27, 28, 42] which have been investigated in other health conditions could be effective and important to meet therapeutic windows; consequently their application (besides pharmacological treatments) needs to be studied in the population of Covid-19 patients, particularly those with severe disease. Third, this involves a better integration of primary care and psychiatric and rehabilitation services which is unfortunately lacking in many lower as well as higher

resourced countries including China [43]. Fourth, systematic community follow up and two-way referral systems between community health centers and specialized hospital units, another weak point of many health systems, need to be strengthened [43]. Fifth, we recommend that future prospective cohort studies in Covid-19 survivors include functional and psychiatric outcomes.

## Conclusion

A significant proportion of Covid-19 survivors had disability and anxiety at discharge from hospital. Disease severity was the only independent risk factor with consistent adverse effects on all outcomes. Health systems need to be prepared for an additional long-term burden due to Covid-19. This includes raising awareness about mental and physical health problems of survivors, early interventions, strengthening medical follow up, and increasing physical and psychological rehabilitation capacity.

## Supporting information

**S1 Appendix. Estimates of risk ratios and 95% confidence intervals from log-linear Poisson regression with robust standard errors.**
(DOCX)

**S1 Checklist. STROBE statement—Checklist of items that should be included in reports of observational studies.**
(DOCX)

## Acknowledgments

We thank all the hospital staff members and coordinators under the CSPRM Covid-19 network for their efforts in collecting the information that was used in this study; Liangjiang Huang (Tongji Hospital, Tongji Medical College, Huazhong University of Science and Technology), Xu Qin (No.3 People's Hospital of Chengdu), Minqing Li (Dazhou Central Hospital), Kehui Hu (Suining Central Hospital), and Jinling Zhang (West China Second University Hospital, Sichuan University) for their dedication to data entry and verification; Prof. Deying Kang (Epidemiologist, West China Hospital, Sichuan University) and Mrs. Rouyue Fang (Statistician, Industrial and Commercial Bank of China) for their assistance in initial data extraction and processing; all the patients who consented to provide their data for analysis and the medical staff members who are at the frontline caring for patients with Covid-19. We would like to thank Prof. Dr. Gerold Stucki and Cristiana Baffone for constructive comments on an earlier version of the manuscript.

## Author Contributions

**Conceptualization:** Siyi Zhu, Qiang Gao, Lin Yang, Wenguang Xia, Chunping Du, Chengqi He, Jan D. Reinhardt.

**Data curation:** Siyi Zhu, Qiang Gao, Lin Yang, Yonghong Yang, Wenguang Xia, Yanping Hui, Chunping Du, Hongchen He, Yun Qu, Chengqi He, Jan D. Reinhardt.

**Formal analysis:** Siyi Zhu, Qiang Gao, Lin Yang, Yonghong Yang, Wenguang Xia, Hongchen He, Yun Qu, Chengqi He, Jan D. Reinhardt.

**Funding acquisition:** Siyi Zhu, Chengqi He, Jan D. Reinhardt.

**Investigation:** Siyi Zhu, Qiang Gao, Lin Yang, Yonghong Yang, Xiguo Cai, Yanping Hui, Di Zhu, Yanyan Zhang, Guiqing Zhang, Shuang Wu, Yiliang Wang, Zhiqiang Zhou, Hongfei Liu, Changjie Zhang, Bo Zhang, Jianrong Yang, Mei Feng, Baoyu Chen, Chunping Du, Hongchen He, Quan Wei, Chengqi He, Jan D. Reinhardt.

**Methodology:** Siyi Zhu, Qiang Gao, Lin Yang, Quan Wei, Chengqi He, Jan D. Reinhardt.

**Project administration:** Siyi Zhu, Yonghong Yang, Wenguang Xia, Xiguo Cai, Di Zhu, Shuang Wu, Mei Feng, Chengqi He, Jan D. Reinhardt.

**Resources:** Siyi Zhu, Lin Yang, Yonghong Yang, Xiguo Cai, Yanping Hui, Di Zhu, Yanyan Zhang, Guiqing Zhang, Shuang Wu, Yiliang Wang, Zhiqiang Zhou, Hongfei Liu, Changjie Zhang, Bo Zhang, Jianrong Yang, Mei Feng, Zhong Ni, Baoyu Chen, Chunping Du, Hongchen He, Chengqi He, Jan D. Reinhardt.

**Software:** Siyi Zhu, Jan D. Reinhardt.

**Supervision:** Siyi Zhu, Yonghong Yang, Yun Qu, Quan Wei, Chengqi He, Jan D. Reinhardt.

**Validation:** Siyi Zhu, Qiang Gao, Lin Yang, Yonghong Yang, Xiguo Cai, Yanyan Zhang, Zhong Ni, Yun Qu, Chengqi He, Jan D. Reinhardt.

**Visualization:** Siyi Zhu, Yonghong Yang, Yanping Hui, Yanyan Zhang, Jan D. Reinhardt.

**Writing – original draft:** Siyi Zhu, Qiang Gao, Yonghong Yang, Yanping Hui, Hongchen He, Chengqi He, Jan D. Reinhardt.

**Writing – review & editing:** Siyi Zhu, Qiang Gao, Lin Yang, Wenguang Xia, Xiguo Cai, Yanping Hui, Di Zhu, Yanyan Zhang, Guiqing Zhang, Shuang Wu, Yiliang Wang, Zhiqiang Zhou, Hongfei Liu, Changjie Zhang, Bo Zhang, Jianrong Yang, Mei Feng, Zhong Ni, Baoyu Chen, Chunping Du, Yun Qu, Quan Wei, Chengqi He, Jan D. Reinhardt.

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
