## [Decision Letter · Decision Letter 0]

23 Sep 2020

PONE-D-20-25960

Prevalence and risk factors of disability and anxiety in a retrospective cohort of 432 survivors of Coronavirus Disease-2019 (Covid-19) from China

PLOS ONE

Dear Dr. Zhu,

Thank you for submitting your manuscript to PLOS ONE. After careful consideration, we feel that it has merit but does not fully meet PLOS ONE’s publication criteria as it currently stands. Therefore, we invite you to submit a revised version of the manuscript that addresses the points raised during the review process.

**Reviewers have taken an interest in the study and found it to be valuable work in several respects. However, several limitations of the manuscript still do not allow a final decision to be made as to whether the study should be published. I suggest that the authors take advantage of the suggestions of the two reviewers to improve the quality of the manuscript and make it suitable for publication.**

We look forward to receiving your revised manuscript.

Kind regards,

Stefano Federici, Ph.D.

Academic Editor

PLOS ONE

Journal Requirements:

2. Please request the authors to provide clinical triage documents.

4.We note that you have indicated that data from this study are available upon request. PLOS only allows data to be available upon request if there are legal or ethical restrictions on sharing data publicly. For more information on unacceptable data access restrictions, please see http://journals.plos.org/plosone/s/data-availability#loc-unacceptable-data-access-restrictions.

5.Thank you for stating the following in the Funding Section of your manuscript:

[This study was funded by the National Natural Science Foundation (81972146), the

Department of Science and Technology of Sichuan Province (20YYJC3320), China

Postdoctoral Science Foundation (2020M673251), Health Commission of Sichuan

Province (20PJ034), and West China Hospital of Sichuan University (HX-2019-nCoV-

011 to Chengqi He, and 2019HXBH058 to Siyi Zhu). The funders played no role in the design, conduct, or reporting of this study.]

 [The funders had no role in study design, data collection and analysis, decision to publish, or preparation of the manuscript.]

Additional Editor Comments (if provided):

Reviewers have taken an interest in the study and found it to be valuable work in several respects. However, several limitations of the manuscript still do not allow a final decision to be made as to whether the study should be published. I suggest that the authors take advantage of the suggestions of the two reviewers to improve the quality of the manuscript and make it suitable for publication.

Reviewers' comments:

Reviewer's Responses to Questions

**Comments to the Author**

1. Is the manuscript technically sound, and do the data support the conclusions?

Reviewer #1: Yes

Reviewer #2: Yes

2. Has the statistical analysis been performed appropriately and rigorously? 

Reviewer #1: I Don't Know

Reviewer #2: No

3. Have the authors made all data underlying the findings in their manuscript fully available?

Reviewer #1: Yes

Reviewer #2: Yes

4. Is the manuscript presented in an intelligible fashion and written in standard English?

Reviewer #1: Yes

Reviewer #2: Yes

5. Review Comments to the Author

Reviewer #1: Overall, I think that this is a useful study and there have been few publications to date that have considered the post-acute aspects of COVID-19. I do feel that the study itself has several limitations, however, which are largely highlighted by the authors. My main issue is the lack of data regarding functional status prior to hospitalisation. The authors have used co-morbidity as a surrogate marker for this but given that hypertension is included as such, I do not think that there would necessarily be a strong correlation with functional dependence so I am not sure how valid it really is to use this. I would presume that the majority of patients admitted to hospital would have some background information gathered on their level of dependency/frailty as it usually forms a standard part of the history taking. However, given that this was a retrospective study and many of the patients were from Hubei province and admitted at the height of the pandemic, these details may not have been recorded and I also recognise that an incomplete data set may not add a huge amount to the interpretation of the findings. It is also interesting that a single recorded respiratory rate of above 30 is considered in isolation to be a marker of severe disease given that there could be many factors that could contribute to this (co-existent COPD or other chronic lung/heart condition/metabolic derangement/inaccurate documentation).

Although self-reported questionnaires may be less accurate than those recorded by the investigators, given that the findings are highly predictable, I suspect that overall would make little difference to the outcomes. I think that the study would be greatly enhanced by reassessing the same cohort at 3 months and it could be argued that this will provide more meaningful information regarding the longer term needs of those who have had COVID-19. However, given that early rehab/intervention programmes have long been established as key for effective rehabilitation, this study highlights the burden of disability at discharge and thus makes an important point regarding resource allocation and need for investment in post-acute COVID services (which in my experience has been somewhat overlooked).

There are a few typos (line 152 appears to be missing a word) and the introduction is slightly ‘media-like’ but overall I think it is reasonably well written and gets the points across well, in particular the limitations

Reviewer #2: This ms presents findings of analyses of data from 424 Covid-19 survivors in 8 provinces of China that make a strong case that those with more severe disease have increased disability and anxiety at discharge from the hospital. There are several areas of concern that the authors need to address before a final decision can be made regarding publication.

1. While the results in the tables and Figures convey findings that are clearly statistically significant – e.g., Risk Ratios and 95%CIs for effects of Disease Severity, Ethnicity and Province on Anxiety in Figure 2 that do not overlap with 1 – it would help the reader appreciate these effects if the authors included P-values in the tables and text – e.g., “Having severe Covid-19 was the strongest risk factor for probable clinically relevant anxiety (P<0.00xx).” It would also be helpful to provide P-values for the age x disease severity effects reported in Table 2 and illustrated in Figure 3.

2. How are the three outcomes of IADL, ADL dependence and Anxiety correlated with each other? It is likely that those with more IADL and/or ADL dependence with have higher Anxiety levels, and it will be good to document and report whether that is the case. There is evidence in the literature that these factors are correlated (e.g., Clancy F, O'Connor DB, Prestwich A. Do Worry and Brooding Predict Health Behaviors? A Daily Diary Investigation. Int J Behav Med. 2020 Oct;27(5):591-601).

3. Given that the three outcomes are likely to be correlated, it could also be informative if the authors performed additional analyses to determine whether, e.g., the effect of disease severity to increase IADL is mediated by the effect of disease severity to increase Anxiety (or vice versa). This could be accomplished by using structural equation modeling to determine whether there is a significant indirect path – i.e., increased disease severity � increased Anxiety � increased IADL.

4. In addition to Anxiety it would have also been informative if the authors had included a measure of depression – e.g., Zung Self-Rating Depression scale – to determine whether disease severity was also associated with the psychological disorder. They cited evidence (ref 31) showing that depression levels are increased in Chinese health Care Workers exposed to coronavirus disease, so it is likely that depression levels are also elevated in the patients with severe disease in the current study.

5. The authors note the important implication of their findings that early psychological/rehabilitation interventions could be useful in improving mental and physical health in Covid-19 survivors. There is evidence that training in cognitive behavioral stress management skills has been effective in reducing depression and anxiety in male cardiac surgery patients in Singapore (Bishop GD, Kaur D, Tan VL, Chua YL, Liew SM, Mak KH. Effects of a psychosocial skills training workshop on psychophysiological and psychosocial risk in patients undergoing coronary artery bypass grafting. Am Heart J. 2005 Sep;150(3):602-9) and Chinese medical students (Li C, Chu F, Wang H, Wang XP. Efficacy of Williams LifeSkills training for improving psychological health: a pilot comparison study of Chinese medical students. Asia Pac Psychiatry. 2014 Jun;6(2):161-9).

6. PLOS authors have the option to publish the peer review history of their article (what does this mean?). If published, this will include your full peer review and any attached files.

Reviewer #1: No

Reviewer #2: No

---

## [Author Response · Author response to Decision Letter 0]

4 Nov 2020

Reviewers' Comments to the Authors: 

Reviewer 1

Overall, I think that this is a useful study and there have been few publications to date that have considered the post-acute aspects of COVID-19. I do feel that the study itself has several limitations, however, which are largely highlighted by the authors.

Author response: Thank you.

1. My main issue is the lack of data regarding functional status prior to hospitalisation. The authors have used co-morbidity as a surrogate marker for this but given that hypertension is included as such, I do not think that there would necessarily be a strong correlation with functional dependence so I am not sure how valid it really is to use this. I would presume that the majority of patients admitted to hospital would have some background information gathered on their level of dependency/frailty as it usually forms a standard part of the history taking. However, given that this was a retrospective study and many of the patients were from Hubei province and admitted at the height of the pandemic, these details may not have been recorded and I also recognise that an incomplete data set may not add a huge amount to the interpretation of the findings.

Author response: The reviewer is correct that this is a limitation which however usually occurs with many health status measures unless people are participants in some large prospective cohort study that routinely gathers data on function before they develop a particular disease. The reviewer is also correct that comorbidity not necessarily correlates strongly with function in every case as other factors such as age and environmental factors may play a role. In the situation within which the patients were admitted However, comorbidity is the best proxy that is available with the current data. As noted by the reviewer, this imitation is discussed by us.

3. It is also interesting that a single recorded respiratory rate of above 30 is considered in isolation to be a marker of severe disease given that there could be many factors that could contribute to this (co-existent COPD or other chronic lung/heart condition/metabolic derangement/inaccurate documentation).

Author response: This is certainly correct. As discussed in the manuscript the criteria for classification of community acquired pneumonia as severe recommended by the American Thoracic Society are more strict. The criteria applied in the present data were the criteria recommended by the Chinese Health Commission at the time of the study. These criteria were applied in Chinese hospitals and they were thus relevant for clinical decision making. Given that little was known about the course of Covid-19 at the time of the study a more liberal definition of severe disease makes sense in order to alert clinicians early.

4. Although self-reported questionnaires may be less accurate than those recorded by the investigators, given that the findings are highly predictable, I suspect that overall would make little difference to the outcomes. I think that the study would be greatly enhanced by reassessing the same cohort at 3 months and it could be argued that this will provide more meaningful information regarding the longer term needs of those who have had COVID-19. However, given that early rehab/intervention programmes have long been established as key for effective rehabilitation, this study highlights the burden of disability at discharge and thus makes an important point regarding resource allocation and need for investment in post-acute COVID services (which in my experience has been somewhat overlooked.

Author response: We agree with the reviewer’s opinion. A protocol and grant proposal for an extended prospective cohort study following these and additional patients has been submitted to the Sichuan Bureau of Science and Technology and is currently under review. Another longitudinal study lead by one of the authors (JR) and funded by the Xi’an Bureau of Technology (XA2020-HWYZ-0043 ) and Xi’an Xidian Hospital on functional and psychological health status of Covid-19 survivors is currently ongoing. Findings will be published after conclusion of these studies which are intended to run over a 12 months period. We find it nevertheless meaningful to report data on functional disability and anxiety at discharge because this highlights two important issues also mentioned by the reviewer: 1) it could be beneficial to integrate early rehabilitation interventions into acute care, 2) many Covid-19 survivors are in need of additional medical services following discharge acute care.

5. There are a few typos (line 152 appears to be missing a word) and the introduction is slightly ‘media-like’ but overall I think it is reasonably well written and gets the points across well, in particular the limitations.

Author response: Thank you for pointing this out. We have thoroughly proofread the manuscript and reworded the sentence on page 7. The introduction is a little ‘media-like’ on purpose: We want to make sure to get the point across that we need health systems to plan for the rehabilitation of Covid-19 survivors.

 

Reviewer 2

This MS presents findings that make a strong case that those with more severe disease have increased disability and anxiety at discharge from the hospital.

Author response: Many thanks!

1. There are several areas of concern that the authors need to address before a final decision can be made regarding publication. While the results in the tables and Figures convey findings that are clearly statistically significant – e.g., Risk Ratios and 95%CIs for effects of Disease Severity, Ethnicity and Province on Anxiety in Figure 2 that do not overlap with 1 – it would help the reader appreciate these effects if the authors included P-values in the tables and text – e.g., “Having severe Covid-19 was the strongest risk factor for probable clinically relevant anxiety (P<0.00xx).” It would also be helpful to provide P-values for the age x disease severity effects reported in Table 2 and illustrated in Figure 3.

Author response: P-values have been added to the tables and effect sizes and confidence intervals or p-values have been integrated into the main text. P-values (if exact values are given) and confidence intervals are essentially equivalent and can be transformed into each other given the unstandardized effect size (see: Altman DG, Bland M. How to obtain the P value from a confidence interval. BMJ 2011, 343: d2304 and Altman DG, Bland M. How to obtain a confidence interval from a P value. BMJ 2011, 342: d2090). In the manuscript p-values are now provided for group comparison (severe vs non severe disease) with chi-squared test, Mann-Whitney U test, or logistic regression with regard to demographics, clinical characteristics and prevalence of outcomes. Risk Ratios and Incidence Rate Ratios with 95% confidence intervals are now provided in the manuscript text as well for the models. P-values have been added to all tables.

2. How are the three outcomes of IADL, ADL dependence and Anxiety correlated with each other? It is likely that those with more IADL and/or ADL dependence with have higher Anxiety levels, and it will be good to document and report whether that is the case. There is evidence in the literature that these factors are correlated (e.g., Clancy F, O'Connor DB, Prestwich A. Do Worry and Brooding Predict Health Behaviors? A Daily Diary Investigation. Int J Behav Med. 2020 Oct;27(5):591-601).

Author response: Information on the association between outcomes is now provided as detailed below.

Methods (Lines 192-195, page 9):

The strength of association between outcomes is reported as Cramer’s V with p-values from chi-squared tests. Cramer’s V ranges from zero (no association) to one (perfect association) with values from 0.3 to 0.5 indicating moderate effects and values above 0.5 indicating large effects.

Results (Lines 240-245, 15):

Associations between outcomes

Outcomes showed moderate to strong associations with Cramer’s V being 0.56 for the association between one or more IADL limitation and at least moderate dependence, 0.39 for the association between one or more IADL limitation and probable clinically relevant anxiety, and 0.33 for the association between at least moderate dependence and anxiety (all p < 0.001).

Discussion (Lines 362-364, page 21)

We found moderate to strong associations between outcomes, alluding to possible processes of mediation, i.e. indirect effects on one outcome triggered by the other outcomes.

3. . Given that the three outcomes are likely to be correlated, it could also be informative if the authors performed additional analyses to determine whether, e.g., the effect of disease severity to increase IADL is mediated by the effect of disease severity to increase Anxiety (or vice versa). This could be accomplished by using structural equation modeling to determine whether there is a significant indirect path – i.e., increased disease severity � increased Anxiety � increased IADL.

Author response: Thanks for this well taken comment. Mediation and possible indirect effects are now discussed. Given the cross-sectional nature of outcomes assessment and the novel and explorative character of this study, we however refrained from conducting complex analyses such as those involving SEM. Furthermore, such hypotheses had not been pre-specified. We discuss the issue under limitations.

Study Limitations (Lines 447-451, page 25)

Fifth, outcomes were correlated with each other pointing at possible mediation of effects. Indirect effects due to mediation were however not formally examined here given the cross-sectional nature of outcome assessment. Furthermore, we refrained from more complex analyses due to the novel character of the presented data and the therefore exploratory character of analysis.

4. In addition to Anxiety it would have also been informative if the authors had included a measure of depression – e.g., Zung Self-Rating Depression scale – to determine whether disease severity was also associated with the psychological disorder. They cited evidence (ref 31) showing that depression levels are increased in Chinese health Care Workers exposed to coronavirus disease, so it is likely that depression levels are also elevated in the patients with severe disease in the current study.

Author response: Thank you for this useful suggestion. We will incorporate this suggestion into the protocol of our upcoming prospective cohort study aimed at long-term follow up on functional and psychological status of Covid-19 survivors.

5. The authors note the important implication of their findings that early psychological/rehabilitation interventions could be useful in improving mental and physical health in Covid-19 survivors. There is evidence that training in cognitive behavioral stress management skills has been effective in reducing depression and anxiety in male cardiac surgery patients in Singapore (Bishop GD, Kaur D, Tan VL, Chua YL, Liew SM, Mak KH. Effects of a psychosocial skills training workshop on psychophysiological and psychosocial risk in patients undergoing coronary artery bypass grafting. Am Heart J. 2005 Sep;150(3):602-9) and Chinese medical students (Li C, Chu F, Wang H, Wang XP. Efficacy of Williams LifeSkills training for improving psychological health: a pilot comparison study of Chinese medical students. Asia Pac Psychiatry. 2014 Jun;6(2):161-9).

Author response: Thank you. We have reviewed the relevant evidence and cited the respective references in the revised manuscript.

---

## [Decision Letter · Decision Letter 1]

1 Dec 2020

Prevalence and risk factors of disability and anxiety in a retrospective cohort of 432 survivors of Coronavirus Disease-2019 (Covid-19) from China

PONE-D-20-25960R1

Dear Dr. Reinhardt,

We’re pleased to inform you that your manuscript has been judged scientifically suitable for publication and will be formally accepted for publication once it meets all outstanding technical requirements.

Kind regards,

Stefano Federici, Ph.D.

Academic Editor

PLOS ONE

Additional Editor Comments (optional):

Reviewers' comments:

Reviewer's Responses to Questions

**Comments to the Author**

1. If the authors have adequately addressed your comments raised in a previous round of review and you feel that this manuscript is now acceptable for publication, you may indicate that here to bypass the “Comments to the Author” section, enter your conflict of interest statement in the “Confidential to Editor” section, and submit your "Accept" recommendation.

Reviewer #2: All comments have been addressed

2. Is the manuscript technically sound, and do the data support the conclusions?

Reviewer #2: Yes

3. Has the statistical analysis been performed appropriately and rigorously? 

Reviewer #2: Yes

4. Have the authors made all data underlying the findings in their manuscript fully available?

Reviewer #2: Yes

5. Is the manuscript presented in an intelligible fashion and written in standard English?

Reviewer #2: Yes

6. Review Comments to the Author

Reviewer #2: The authors have responded well to my concerns. There's only a minor problem they need to address -- i.e., net to insert P<0.001 next to "Pre-existing comorbidity" in Table 1

7. PLOS authors have the option to publish the peer review history of their article (what does this mean?). If published, this will include your full peer review and any attached files.

Reviewer #2: No

---

## [Editor Report · Acceptance letter]

10 Dec 2020

PONE-D-20-25960R1 

Prevalence and risk factors of disability and anxiety in a retrospective cohort of 432 survivors of Coronavirus Disease-2019 (Covid-19) from China 

Dear Dr. Reinhardt:

I'm pleased to inform you that your manuscript has been deemed suitable for publication in PLOS ONE. Congratulations! Your manuscript is now with our production department. 

Kind regards, 

on behalf of

Prof. Stefano Federici 

Academic Editor

PLOS ONE